# Energy Consumption Patterns and Load Forecasting with Profiled CNN-LSTM Networks

**Kareem Al-Saudi \*** **, Viktoriya Degeler** and **Michel Medema**

Bernoulli Institute for Mathematics, Computer Science and Artificial Intelligence, University of Groningen, 9747 AG Groningen, The Netherlands; V.Degeler@rug.nl (V.D.); M.Medema@rug.nl (M.M.)
\* Correspondence: KareemAlSaudi13@gmail.com

**Abstract:** By virtue of the steady societal shift to the use of smart technologies built on the increasingly popular smart grid framework, we have noticed an increase in the need to analyze household electricity consumption at the individual level. In order to work efficiently, these technologies rely on load forecasting to optimize operations that are related to energy consumption (such as household appliance scheduling). This paper proposes a novel load forecasting method that utilizes a clustering step prior to the forecasting step to group together days that exhibit similar energy consumption patterns. Following that, we attempt to classify new days into pre-generated clusters by making use of the available context information (day of the week, month, predicted weather). Finally, using available historical data (with regard to energy consumption) alongside meteorological and temporal variables, we train a CNN-LSTM model on a per-cluster basis that specializes in forecasting based on the energy profiles present within each cluster. This method leads to improvements in forecasting performance (upwards of a 10% increase in mean absolute percentage error scores) and provides us with the added benefit of being able to easily highlight and extract information that allows us to identify which external variables have an effect on the energy consumption of any individual household.

**Keywords:** pattern recognition; energy profiling; clustering; forecasting



## 1. Introduction

Over the years, our reliance on electrical appliances has been slowly increasing. As our dependence on electrical appliances has increased, so too has our consumption of energy [1] and, subsequently, our need for more sophisticated and advanced solutions that can accommodate this growth. Thankfully, the convergence of multiple technologies—such as machine learning, data mining and ubiquitous computing—has led to the rise of a solution in the form of smart (electric) grids as well as smart environments and smart meters, which are slowly but surely taking off in terms of their popularity and availability [2]. The resulting growth in the prevalence of smart grids has given us the opportunity to both control and monitor the energy consumption of individual households on a real-time basis [3], and, through the utilization of applications built using this framework, we are capable of achieving an overall reduction in the amount of energy that we, as the human race, consume. This opens up the possibility to alleviate some of the inherent risks associated with the growth in energy consumption, whether that be our overall environmental footprint on the planet or, on a much smaller scale, the financial impact on both suppliers as well as consumers due to instabilities present in current, outdated power grid systems [4].

Existing solutions developed under the increasingly popular smart grid framework, such as the Home Energy Management System (HEMS) and Battery Energy Management System (BEMS), aim to provide the end-user with the means to schedule, or otherwise manage, daily appliance operations, taking into consideration external factors such as weather conditions and utility tariff rates alongside any other personal preferences [3].

To operate efficiently, these solutions rely on our ability to capably forecast future trends in energy consumption at the individual household level. This information is required to allow appropriate and sufficient control and to supply the correct energy load to the end-user [5,6]. This has lead to a shift in interest within the realm of load forecasting, whereby research has moved from a predominant focus at the large-scale, regional level [7], where an amalgamation of available data spanning numerous households provides more obvious patterns as a result of the underlying diversity between households being lost [8], towards a focus at the individual household level. Furthermore, owing to the operational characteristics of both HEMS and BEMS and similar applications, load forecasting in the very short term (anywhere from a few minutes to a couple of hours), which is oftentimes referred to as very short-term load forecasting (VSTLF), is more relevant than the substantially studied longer term horizons that are predominantly associated with long-term network planning and operations [3].

When exploring energy consumption at the individual household level, the diversity and complexity associated with human behavior leads to extremely dynamic, volatile patterns that can prove to be highly dissimilar between households. In addition to this, certain households exhibit no clear pattern in energy consumption due to a high level of irregularity in the lifestyles of their occupants [8]. To account for this dissimilarity, current state-of-the-art methods benefit from a precursory clustering step within the forecasting pipeline [3,4,8]. This precursory clustering step serves to amalgamate days that exhibit a measure of similarity in terms of their energy consumption patterns into the same cluster. By training individual forecasting models on a per-cluster basis, we should, in theory, see an improvement in the load forecasting performance, as each of the respective models specializes in predicting future trends in energy consumption based on patterns present within the energy profile associated with its unique cluster. This is the area of research that this paper seeks to tackle—how can we best construct energy profiles out of historical data that truly capture repeated patterns with regards to energy consumption, and what are the effects of including a clustering step on the performance of a forecasting pipeline.

To attempt to solve the previously outlined VSTLF problem at the individual household level, we propose a novel solution that utilizes a combination of statistical knowledge and machine learning techniques to generate energy profiles that provide us with some measure of insight into the habits of a household's occupants as well as forecast future trends in energy consumption. A high-level overview of the steps pertaining to our proposed model is presented in Figure 1.

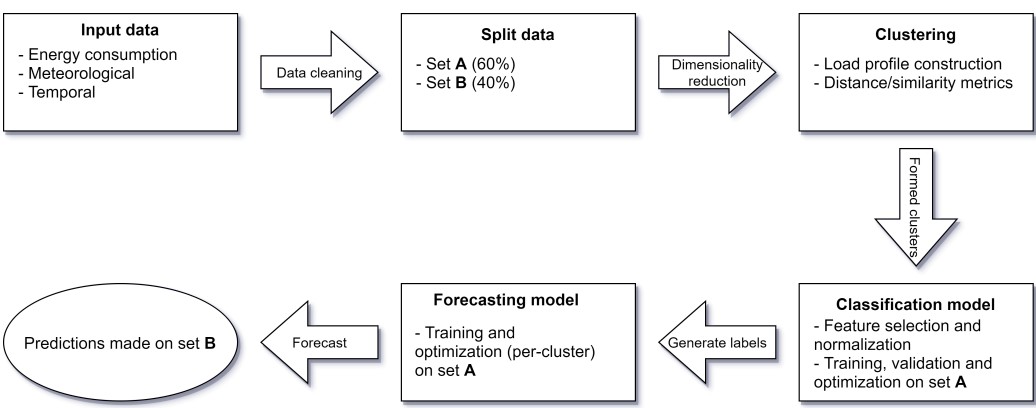

**Figure 1.** High-level overview of the steps pertaining to our model.

To summarize, our method consists of three steps: cluster, classify, and forecast. First, historical days are clustered based on their similarity in terms of active power consumption. Following that, new days are classified into one of the generated clusters, and finally, forecasts are generated based on models that are trained on a per-cluster basis. Our solution takes advantage of modern techniques and algorithms, such as the uniform manifold ap-

proximation and projection algorithm (an extension to the well-received t-SNE algorithm) as well as hierarchical density-based clustering algorithms and deep learning models, which combine aspects of both convolutional neural networks as well as long short-term memory networks, in an attempt to achieve improved forecasting accuracy results when considering what has already been researched in the current literature. Furthermore, we hope to gain some insight into energy consumption habits at the individual household level by constructing unique energy profiles that outline factors that may influence energy consumption. Our contributions to this field are based on measurement of the performance observed when applying our selection of algorithms and networks to publicly available datasets that contain information regarding the active power consumption of individual households (out of the two datasets on hand, one contains directly comparable results [9,10]).

## 2. Related Work

Energy management systems, such as the previously introduced HEMS and BEMS, are designed with the intent to both optimize and control the smart grid energy market. As previously stated, to be able to do this, these demand-side management systems require a priori knowledge about the load patterns, and, as a result of this, the field of designing computationally intelligent load forecasting systems has expanded quite rapidly in recent years with over 50 research papers related to the subject having been identified in existing literature [11]. In this chapter, we explore a compiled subset of this literature that specifically tackles the problem of energy profile construction as well as load forecasting. This is done to establish a baseline for understanding what has already been done within the field in terms of the two focal points of our forecasting pipeline: the precursory clustering step and the state-of-the-art forecasting models. Furthermore, by doing so, we are able to position our paper with respect to the current state-of-the-art models and highlight the key differences in our approach.

### 2.1. Clustering and Energy Profile Creation

The main issue that this paper seeks to address is that of creating interesting profiles in terms of recurrent patterns of energy consumption. To do this, we make use of clustering algorithms that seek to partition our data into a number of clusters so that each of these clusters exhibits some metric of similarity or *goodness*. However, a measure of goodness can inherently be seen as quite subjective with Backer and Jain [12] noting that, "in cluster analysis, a group of objects is split up into a number of more or less homogeneous subgroups on the basis of an often subjectively chosen measure of similarity (i.e., chosen subjectively based on its ability to create "interesting" clusters), such that the similarity between objects within a subgroup is larger than the similarity between objects belonging to different subgroups." We explore papers in the existing literature that present different takes in terms of how they define similarity as well as in their chosen clustering methodologies.

Kong et al. [8] attempted to justify observations made by Stephen et al. [13] by using a density-based clustering technique known as Density Based Spatial Clustering of Applications with Noise (DBSCAN) [14] to evaluate consistency in short-term load profiles. They reported the benefits of using DBSCAN, stating that, as it does not require the number of clusters in the data to be known ahead of time and as it contains the notion of outliers, it is an ideal clustering technique to identify consumption patterns that repeat with a measure of noise akin to what is loosely defined by Practice Theory. Their findings are that the number of clusters as well as the number of outliers vary greatly between households, with some households exhibiting no clearly discernible patterns and some households (mostly) following fairly consistent daily profiles.

Yildiz et al. [3] expanded on traditional load forecasting techniques, such as the Smart Meter Based Model (SMBM), that they had previously presented [15] and presented their own take in the form of a Cluster-Classify-Forecast (CCF) model. In traditional SMBMs, a chosen model, whether a statistical variant or one from the plethora of existing

machine learning models, learns the relationship between target forecasted loads when presented with some input data which, in our case, consist of some historical lags in terms of energy consumption, weather data, and temporal information such as the time and calendar date. The CCF takes this a step further by making use of both K-means and Kohonen's Self-Organizing Map [16] to identify group profiles that are most similar to each other. After obtaining and validating the output of their chosen clustering techniques, the relationship between the clustering output and other temporal variables, such as the weather, is investigated by using a Classification and Regression Tree [17].

### 2.2. Forecasting Models

Numerous studies have been conducted with the intent to forecast energy consumption using methods ranging from multiple linear regression, as assessed by Fumo and Rafe Biswas [18], to the novel deep pooling Recurrent Neural Network introduced by Shi et al. [19]. The majority of these forecasting models, whether statistical, machine learning, or deep learning based, can be classified into two main categories: single technique models, in which only a single, heuristic algorithm (e.g., a Multi-Layer Perceptron or Support Vector Machine) is used as the primary forecasting method, and hybrid methods that encapsulate two or more algorithms [11], such as the Convolutional Neural Network Long Short-Term Memory (CNN-LSTM).

Kong et al. [8] employed a Long Short-Term Memory (LSTM) network, as it is generally ideal when attempting to learn temporal correlations within time series datasets; however, their final results were not very promising, boasting a mean absolute percentage error (MAPE) of approximately 44% over variable time steps. This could have been a result of poor hyperparameter tuning, as they stated that, *"tuning 69 models for each of the candidate methods is very time-consuming for this proof-of-concept paper"*. This leads us to believe that there is definitely room for improvements to be made on the core concepts of their work.

Yildiz et al. [3] used the clusters they formed, as previously described, alongside their assignments to build SMBMs through the use of a Support Vector Regression model, and they found that, alongside improvements to the load forecasting accuracy, they were able to reveal vital information about the habitual load profiles of the households they were exploring. Unfortunately, they do not indicate any potential reasoning as to why they chose to use K-means and Kohonen's SOMs in place of potentially more effective clustering methods, citing only that K-means is the most popular clustering technique [17] and that SOMs is generally used as an extension to neural networks for the purposes of clustering. Additionally, their results only include values that are indicative of their chosen technique's performance on their specific dataset. They presented performance metrics such as the normalized root mean square error (NRMSE) and normalized mean absolute error (NMAE), rendering us unable to compare the performance of their proposed method.

Kim and Cho [9] presented a more modern take on load-forecasting by proposing a hybrid CNN-LSTM network that is capable of extracting both temporal and spatial features present in the data. The use of convolutional layers within the realm of load forecasting is brilliant as it allows the network to take into account the correlations between multivariate variables while minimizing noise that can eventually be fed into the LSTM section of the network that finally generates predictions. Their paper proposes such a network and cites that the major difficulties with such an approach mainly boil down to hyperparameter tuning, which can be remedied through a variety of means including genetic algorithms or packages such as Keras Tuner, which is maintained by O'Malley et al. [20]. Furthermore, Kim and Cho [9] did not explore the possibility of implementing a precursory clustering step which could have lead them to substantial improvements in their final MAPE.

## 3. Dataset Description

At our disposal are a number of publicly available datasets that contain historical energy consumption data. These include data collected by the Engineering and Physical Sciences Research Council via the project entitled "*Personalised Retrofit Decision Support*

*Tools for UK Homes using Smart Home Technology (REFIT)*" [21], which is a collaboration among the Universities of Strathclyde, Loughborough and East Anglia, as well as the *"Individual Household Electric Power Consumption"* dataset [10], which is part of the University of California, Irvine Machine Learning Repository and which is henceforth acronymized as the *"UCI data set (UCID)"*. This section briefly describes the main aspects of each of these individual dat sets so that we may be better able to draw comparisons between them and highlight any key differences. Additionally, we aim to append meteorological features (e.g., temperature, wind speed, cloud coverage, precipitation) to each of our respective datasets. An overview of this process and the data utilized is also presented in this section.

### 3.1. REFIT

The REFIT Electrical Load Measurements dataset [21] includes cleaned electrical consumption data, in watts, for a total of 20 households labeled *House 1- House 21* (skipping House 14), located in Loughborough, a town in England, over the period of 2013 through early 2015. The electrical consumption data were collected at both the aggregate leveland the appliance level, with each household containing a total of 10 power sensors comprising a current clamp for the household aggregate, labelled *Aggregate* in the dataset, as well as nine individual appliance monitors (IAM)m labelled *Appliance 1-Appliance 9* in the dataset. The appliance list associated with each of the IAMs differs between households and includes a measure of ambiguity, as applicants may have switched appliances around during the duration of the data collection and because the installation team responsible for setting up the power sensors did not always collect relevant data associated with the IAMs. The consequences of this are, of course, that we do not know with 100% certainty whether an appliance or set of appliances associated with an IAM remained the same throughout the entirety of time covered by the dataset. Additionally, some labels are inherently ambiguous, for example, the *television site* label, which could includeany number of appliances including televisions, DvD players, computers, speakers, etc. Finally, the makes and models of the appliances that were meant to be collected by the installation team are not always present, further compounding the previously mentioned uncertainties.

The documentation associated with the dataset states that active power was collected and subsequently recorded at intervals of 8 s; however, a cursory glance at the data demonstrates that this was not always the case. A potential reason for this could be the fact that the aforementioned power sensors were not synchronized with the associated collection script which polls within a range of 6 to 8 s, leaving a margin for error in the intervals between recorded data samples. Moreover, the dataset is riddled with long periods of missing data, making it exceptionally difficult to work with. All of that said, the data collection team made an attempt to pre-process or otherwise *clean* the dataset by

1. Correcting the time to account for daylight savings in the United Kingdom;
2. Merging timestamp duplicates;
3. Moving sections of IAM columns to correctly match the appliance they were recording when that appliance was reset or otherwise moved;
4. Forward filling NaN values or zeroing them depending on the duration of the time gap;
5. Removing spikes of greater than 4000 watts from the IAM values and replacing them with zeros;
6. Appending an additional issues columns that was set to 1 if the sum of the sub-metering IAMs was greater than that of the household aggregate—in this case, data should either be discarded or, at the very least, the discrepancy must be noted.

### 3.2. UCID

The UCID dataset [10] contains a total of 2,075,259 measurements gathered from a single house located in Sceaux, a commune in the southern suburbs of Paris, France. The data within this dataset were recorded over a duration of 47 months spanning the period between December 2006 and November 2010. Measurements were made approximately

once per minute and consisted of the minute-averaged active power consumption, in kilowatts, within the entire household as well as three energy submetering measurements, corresponding to the kitchen, which included a dishwasher and microwave; the laundry room, which included a washing machine and tumble dryer; and the combination of both an electric water-heater and an air-conditioner. The UCID dataset is not without fault either, containing approximately 25,979 missing measurements, which make up roughly 1.25% of the entire dataset; however, given the extensive range covered as well as the immense number of total measurements available, these missing values could easily be disregarded and subsequently discarded during the preprocessing stage of our forecasting pipeline.

*3.3. Meteorological Data*

As an addendum to both the REFIT and UCID datasets, we incorporated meteorological data provided by Solcast [22], a company based in Australia that aims to provide high quality and easily-accessible solar data. For the purpose of this master's thesis project, we requested meteorological data with variable time resolutions (5, 10, 15 min) for both the Loughborough area in the United Kingdom for the REFIT dataset and the Sceaux commune in the southern suburbs of Paris, France for the UCID dataset. The relevant periods were 16th September 2013 up to and including 11 July 2015 and 1 December 2006 up to and including 30 November 2010 for each dataset, respectively. The provided data was extensive, covering a wide range of parameters that are listed and described in detail in Table 1.

**Table 1.** List of meteorological parameters available to us from the Solcast datasets.

| Parameter | Description |
|---|---|
| Air Temperature | The air temperature (2 m above ground level). Units are degrees Celsius. |
| Albedo | Average daytime surface reflectivity of visible light, expressed as a value between 0 and 1. 0 represents complete absorption. 1 represents complete reflection. |
| Azimuth | The angle between a line pointing due north to the sun's current position in the sky. Negative to the East. Positive to the West. 0 at due North. Units are degrees. |
| Cloud Opacity | The measurement of how opaque the clouds are to solar radiation in the given location. Units are percentages. |
| Dewpoint | The air dewpoint temperature (2 m above ground level). Units are degrees Celsius. |
| Direct Normal Irradiance | Solar irradiance arriving in a direct line from the sun as measured on a surface held perpendicular to the sun. Units in $W/m^2$. |
| Direct (Beam) Horizontal Irradiance | The horizontal component of Direct Normal Irradiance. Units are $W/m^2$. |
| Global Horizontal Irradiance | The total irradiance received on a horizontal surface. It is the sum of the horizontal components of direct (beam) and diffuse irradiance. Units are $W/m^2$. |
| Global Tilted Irradiance–Fixed | The total irradiance received on a surface with a fixed tilt. The tilt is set to latitude of the location. Units are $W/m^2$. |
| Global Tilted Irradiance–Horizontal Single-Axis Tracker | The total irradiance received on a sun-tracking surface. Units are $W/m^2$. |
| Precipitable Water | The total column preciptable water content. Units are $kg/m^2$. |
| Relative Humidity | The air relative humidity (2 m above ground level). Units are percentages. |
| SFC pressure | The air pressure at ground level. Units are hPa. |
| Snow Depth | The snow depth liquid-water-equivalent. Units are cm. |
| Wind Direction | The wind direction (10 m above ground level). This is the meteorological convention. 0 represents northerly wind (from the north); 90 represents easterly wind (from the east); 180 represents southerly wind (from the south); 270 represents westerly wind (from the west). Units are degrees. |
| Wind Speed | The wind speed (10 m above ground level). Units are m/s. |
| Zenith | The angle between a line perpendicular to the earth's surface and the sun (90 deg = sunrise and sunset; 0 deg = sun directly overhead). Units are degrees. |

## 4. Methodology

This paper proposes a forecasting method that utilizes dimensionality reduction and clustering techniques to group days that exhibit similarities in terms of electric consumption behavior. Days that are grouped into the same cluster are thought to contain shared features, whether those features be temporal, as seen in Table 2, or meteorological or otherwise. The formed clusters (per household) were used for 2 purposes in this study: firstly, to train a classification model that utilizes available context information to assign a new day to the correct cluster and secondly, for the application of a novel deep learning method on a per-cluster basis to forecast future energy consumption. A detailed overview of the proposed model can be seen in Figure 2.

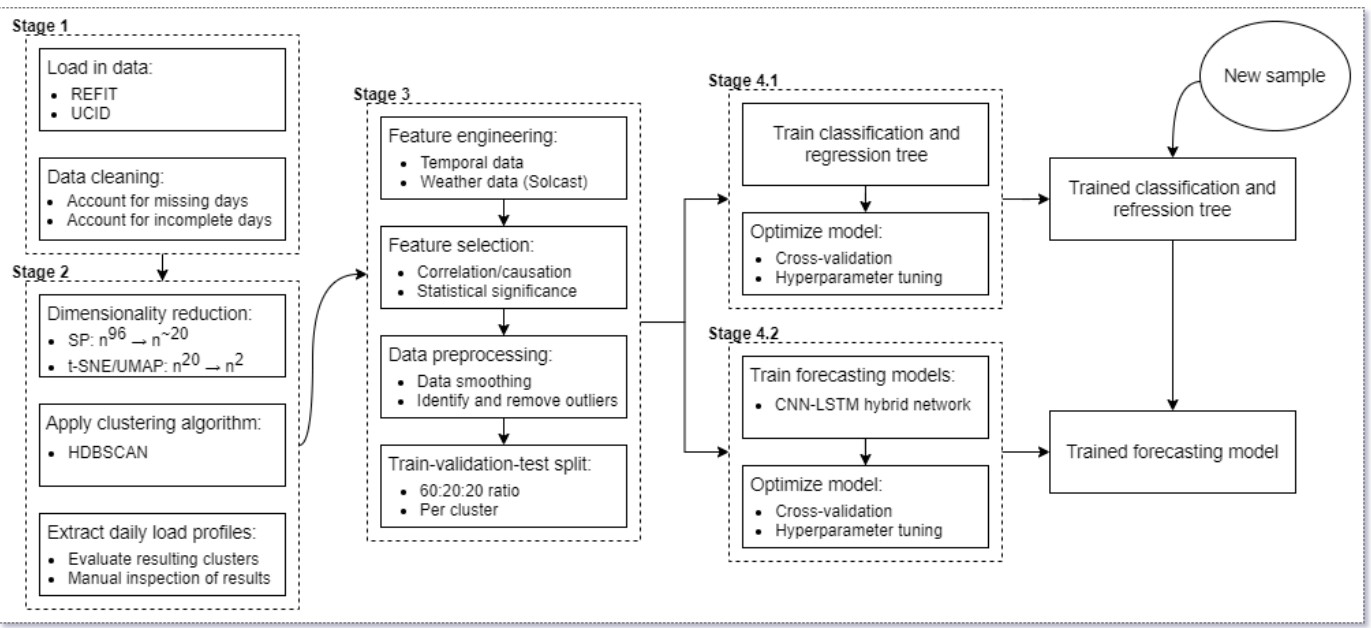

**Figure 2.** Proposed daily profile extraction and load forecasting model.

In short, we started off by resampling the data present in both the REFIT dataset and the UCID dataset into a common resolution so that we could directly compare results. We resampled both datasets to a common resolution of 15 min per sample, as this lined up well with the native resolution of the meteorological data that was provided to us by Solcast. Following this, we cleaned both datasets and ridded them of any days containing an incomplete number of records (here, incomplete refers to any days containing less than 96 records given that we split each day into a total of 96 chunks). After this, we took a subset of each of our datasets (60% of the total data for the REFIT and UCID datasets, henceforth referred to as *Set A*) and left out the remaining 40% (henceforth referred to as *Set B*) in order to validate the results of our forecasting model. We then reduced the overall dimensionality of a single day, going from a total of 96 features to a much more manageable 2 through a combination of statistical and machine learning techniques and generated clusters based on the new, 2-dimensional data set by utilizing a density-based clustering algorithm. Following this, we trained and optimized a classifier on Set A and used it to generate cluster labels for the previously withheld Set B. Finally, we trained our forecasting models, one per cluster, on the data present in Set A and used the data present in Set B to act as both a validation set as well as a training set from which we obtained the final results regarding the forecasting accuracy.

### 4.1. Stage 1— Data Collection and Cleaning

As mentioned previously, this study utilized available historical energy consumption data on an individual household basis. As part of stage 1 of our forecasting pipeline, time series data on daily electricity consumption needed to be collected from an individual household meter for an adequate amount of time at an ideal resolution to obtain acceptable results. After collection of, or in our case, loading in, the data, we performed common preprocessing techniques to account for noisy or otherwise missing data that occurred during the transmission of the data from the meters. The available data were resampled into a resolution of 15 min, and any days that contained less than 96 values (given that days were divided into 96 intervals of 15 min) were dropped from our dataset. All other days that contained NaN values were also not considered and were subsequently dropped from our dataset.

### 4.2. Stage 2— Dimensionality Reduction and Clustering

Given that each day in our dataset was represented by 96 dimensions, with each dimension comprising the mean active power consumption over a time period of 15 min, the first logical step to undertake was to transform the data in a manner that enabled our clustering techniques to more efficiently determine which days exhibited similarities in terms of electric consumption behavior. This *dimensionality reduction* step comprised 2 parts. To start things off, we divided each day into 5 different periods (as per the work of Yildiz et al. [3]), as follows:

1. Morning: 06:00–11:00;
2. Late morning/afternoon: 11:00–15:00;
3. Late afternoon/early evening: 15:00–20:30;
4. Evening: 20:30–23:30;
5. Late evening/early morning: 23:30–06:00.

Following that, we represented each period by its respective mean, and minimum, and maximum values as well as its standard deviation. The outcome of performing this was that each day was represented by a total of 20 dimensions, rather than the initial 96, a reduction of $\sim 80\%$. We were able to reduce this even further, and even visualize our data in 2 or 3 dimensions, by making use of either the t-Distributed Stochastic Neighbor Embedding (t-SNE) [23] or Uniform Manifold Approximation and Projection (UMAP) [24] algorithms. The most important hyperparameter to tune for either algorithm is the *perplexity* hyperparameter for the t-SNE algorithm and the equivalent $n_{\text{neighbors}}$ hyperparameter for the UMAP algorithm. During our research, we found that the optimal value for either of these hyperparameters was $N^{\frac{1}{2}}$, where N is the number of samples present in the dataset.

After performing the dimensionality reduction step on our data, we proceeded to cluster the resulting output by applying the Hierarchical Density Based Spatial Clustering of Applications with Noise (HDBSCAN) algorithm—a hierarchical, non-parametric clustering algorithm proposed by Campello et al. [25] that was designed to overcome the main limitations of DBSCAN. The most substantial changes between HDBSCAN and DBSCAN come in the form of no longer explicitly needing to predefine a value for the distance threshold $\epsilon$. Instead, HDBSCAN generates a complete, density-based clustering hierarchy over variable densities from which a simplified hierarchy composed of only the most significanat clusters in the data can be extracted. The only important parameters that need to be passed to the HDBSCAN algorithm are the minimum size that each cluster is expected to be. In this case, we set that value to $\frac{1}{10}(N)$, where N is the number of samples present in the dataset. Our reason for selecting this value was predominantly based on the adequate results observed by Kong et al. [8] in their implementation of the DBSCAN algorithm in a similar setting whilst utilizing a similar selection in terms of hyperparameter settings. The other hyperparameter that we chose to tune was the *min_samples* hyperparameter, which, in layman's terms, denotes how conservative we would like to be with our clustering in terms of restricting clusters to progressively more dense areas and classifying samples from our dataset as noise. In our case, an arbitrary value of 15 was selected. This was in contrast to the default value that sets *min_samples = min_cluster_size*. The resulting clusters can be found in Section 5 of this paper.

### 4.3. Stage 3— Data Preprocessing

The first step undertaken (on a per-cluster basis) was to append both temporal data and meteorological data to our datasets. Table 1 (located in Section 3.3) pertains to the historical meteorological data concerning the regions associated with our datasets, while Table 2 pertains to the temporal variables that were taken into consideration as part of this feature engineering step. Incidentally, as outlined in Table 2, the temporal variables we have chose to append did not hold much value given their current format. This was due to their cyclical nature (think of how the 23rd hour of the day is rather close to hours 0 and

1). To handle this, we encoded all temporal variables through the use of both the sine and cosine functions in an attempt to transpose our linear interpretation of time into a cyclical state that could be better interpreted by our deep learning model. The result of performing this so-called encoding can be seen in Figure 3a,b, which illustrates a linear-to-cyclical encoding of the time of day (as an example).

**Table 2.** List of temporal variables that were taken into consideration during the feature engineering process, as outlined in Section 4.3.

| Variable | Description |
|---|---|
| Day | An integer value between 1 and 31. |
| Weekday | An integer value between 0 and 6 denoting the different days of the week. |
| Month | An integer value between 1 and 12. |
| Year | An integer value between 2007 and 2010. |
| Hour | An integer value between 0 and 23. |
| Minute | An integer value between 0 and 45 in increments of 15. |
| Season | An integer value between 0 and 3 where 0 denotes Spring, 1 denotes Summer, 2 denotes Fall, and 3 denotes Winter. |
| Holiday | A categorical variable that takes on an integer value of 1 when the day concerned is a public holiday and 0 otherwise. |

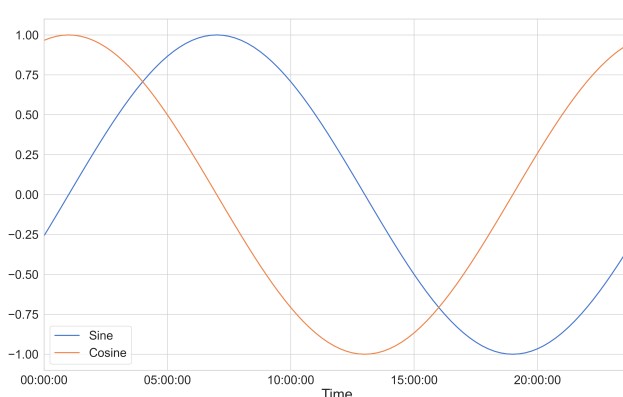

**(a)** The time of the day represented as a combination of both sine and cosine waves.

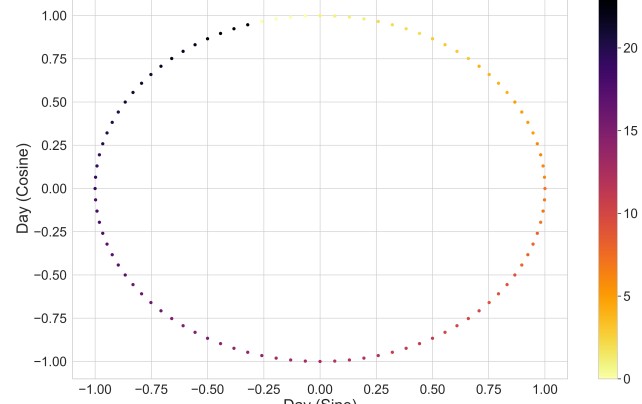

**(b)** Visualizing our cyclical encoding of the time of day.

**Figure 3.** By utilizing a combination of the sine function and the cosine function, we eliminated the possibility that two different times would receive the same value had we used either function independently. The combination of both functions can be thought of as an artificial 2-axis coordinate system that represents the time of day.

Following the feature engineering process, the feature selection process heavily revolved around minimizing the overall number of features that did not serve as good predictors of our target variable. To assess which features should be kept and dropt, we performed a variety of tests to determine which of our features presented a significant level of independent (or combinatorial) correlation or causation when considering our target variable for the REFIT and UCID datasets. The primary tests conducted revolved around the concepts of Granger Causality (see Figure 4) and mutual information gain (see Figure 5), although other factors (such as a per-variable variance threshold) were also looked into.

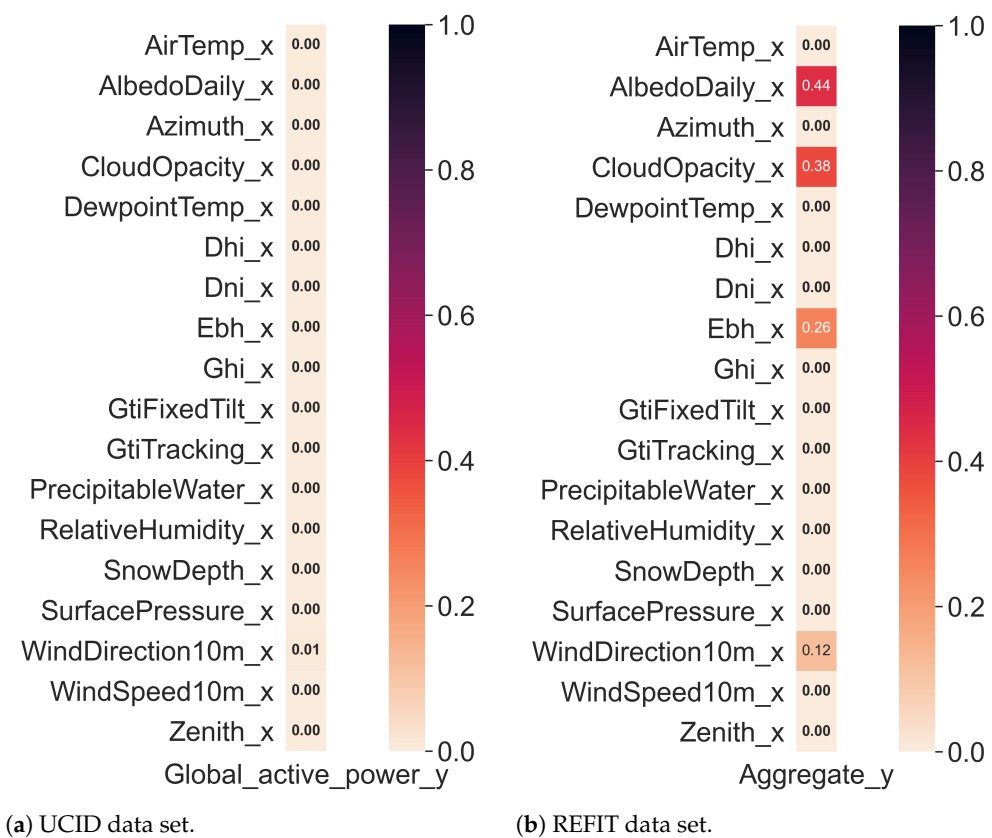

(**a**) UCID data set.   (**b**) REFIT data set.

**Figure 4.** Trimmed Granger Causation matrix that displays the Granger Causality of our independent features against our target variable for the UCID and REFIT datasets.

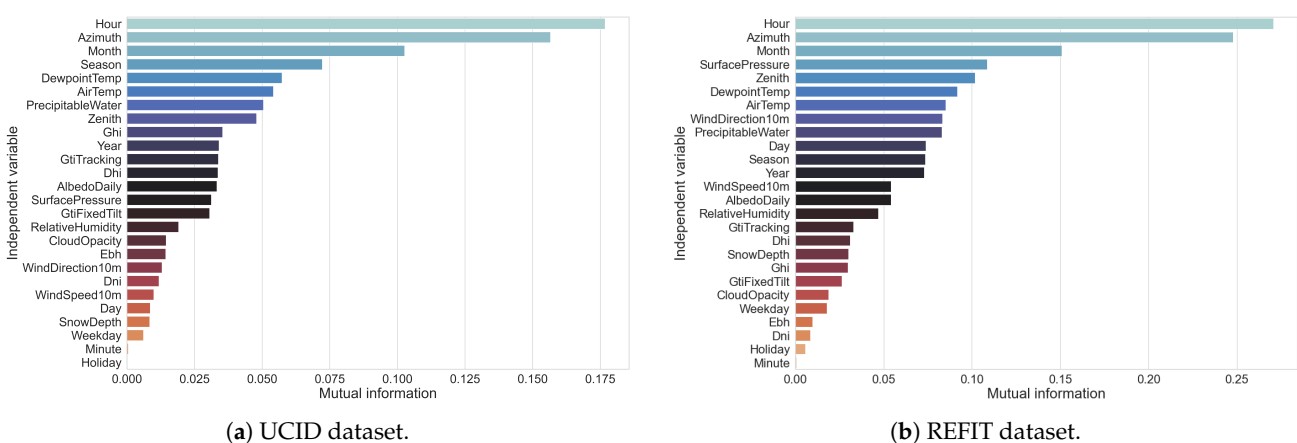

(**a**) UCID dataset.   (**b**) REFIT dataset.

**Figure 5.** Mutual information gain with regard to our independent features and target variable for the UCID and RE-FIT datasets.

When taking our target variable into consideration, the notion of outliers (and how to deal with them) was inevitable. Leaving them in was one possibility, as some level of noise is unavoidable in the data collection process, and training our models on unrealistically curated data would not serve to produce an accurate representation of a real-life scenario in which a model of this caliber could be applied. Alternatively one method, explored during prior, related research [6], works on the basis of defining upper and lower bounds based on the interquartile range (IQR). The IQR is calculated as the difference between the 75th (Q3) and 25th (Q1) percentiles of the data and comprises the box in a traditional box

and whiskers plot. Using the IQR, outliers can be defined as any values that are predefined factors below the 25th percentile or above the 75th percentile, as follows:

$$Q1 - (1.5 * IQR) < x < Q3 + (1.5 * IQR). \tag{1}$$

Figure 6a,b represents the distribution of values for our target variable over the different months of the year before and after removing outliers.

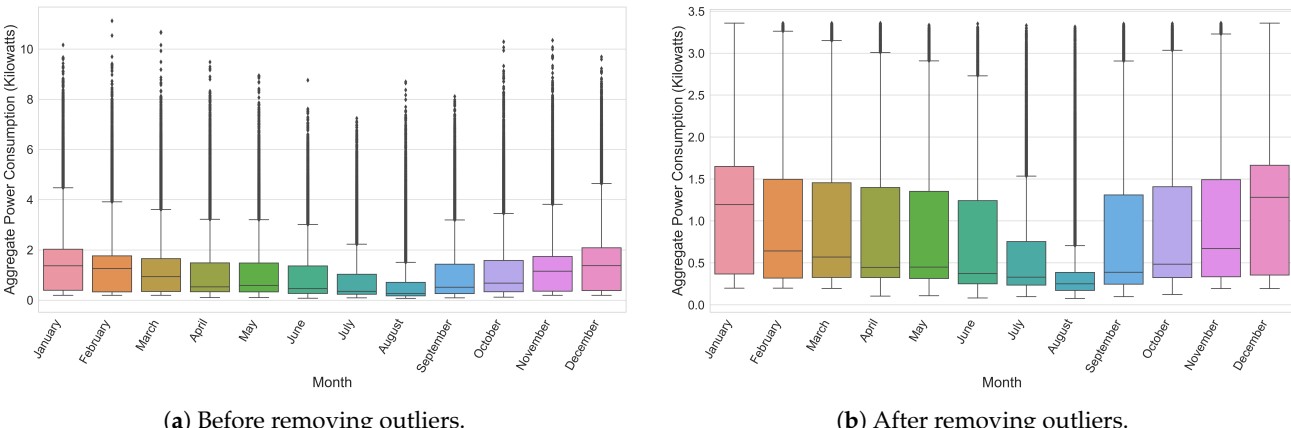

(**a**) Before removing outliers.      (**b**) After removing outliers.

**Figure 6.** Illustration of the distribution of values with respect to the global active power of the UCID dataset before and after removing outlier values, as defined by Equation (1).

*Smoothing* or filtering the data can also be done through the use of a variety of techniques and can help to alleviate some of the issues inherent to the noise present in data as a byproduct of the data collection process. An example of performing a preliminary smoothing step on energy consumption data can be seen in the work of Hsiao [4], in which a moving (or rolling) average method was utilized.

With regard to our proposed forecasting pipeline, we utilized Savitzky–Golay filters [26] to smooth our raw electrical energy consumption data as, when compared to the moving average method, Savitzky–Golay filters tend to do a better job at preserving the integrity of raw data. Figure 7a,b serves to illustrate the application of both the moving average method as well as the Savitzky–Golay filter method on a subset of our (raw) dataset in order to better visualize the differences between the methods.

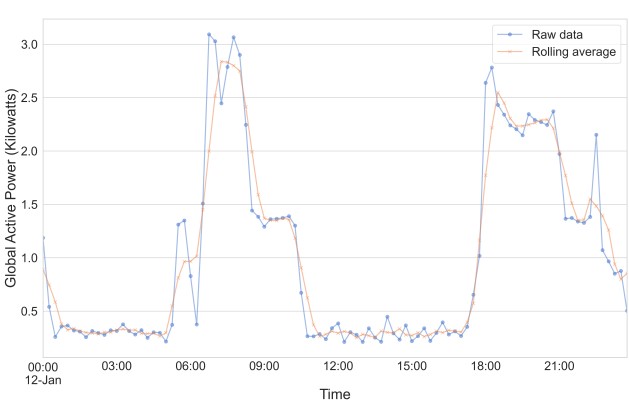 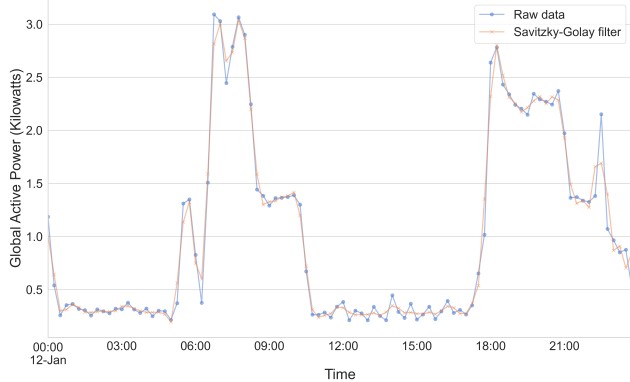

(**a**) Application of the moving average method with a window size of 3.

(**b**) Application of the Savitzky–Golay filter method with a polynomial order of 3 and a window size of 5.

**Figure 7.** Illustration of the application of the moving average method and the Savitzky–Golay filter method to smooth a subset of our raw data.

When considering the trend component of our data (obtained through performing an additive time series decomposition step, as seen in Figure 8), a preliminary smoothing step can be undertaken through the use of Locally Weighted Scatterplot Smoothing (LOESS). This is illustrated in Figure 9.

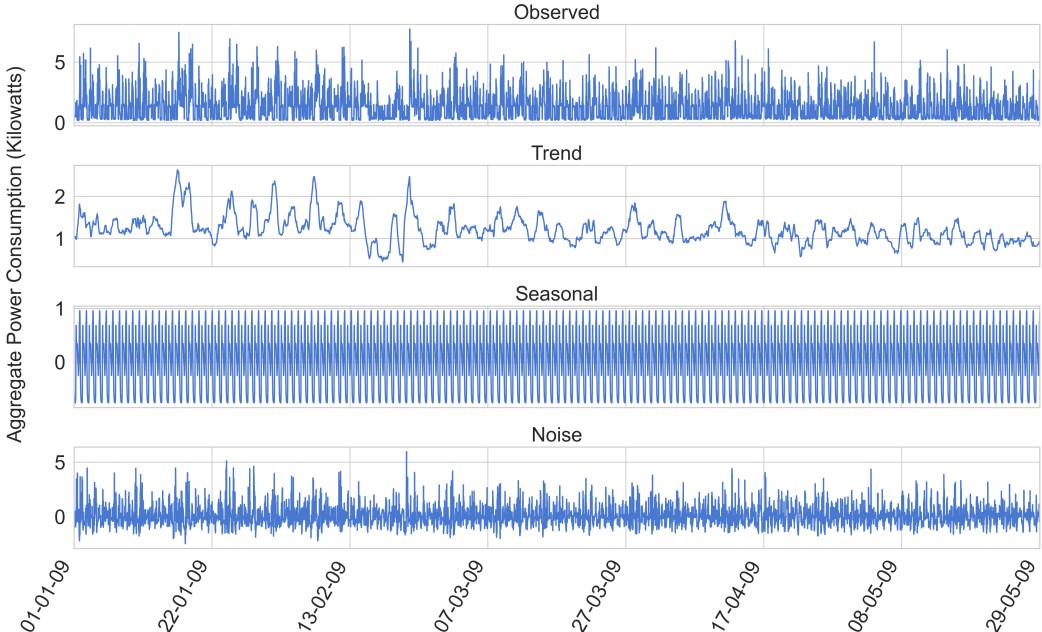

**Figure 8.** The outcome of performing time series decomposition on a subset of the UCID dataset.

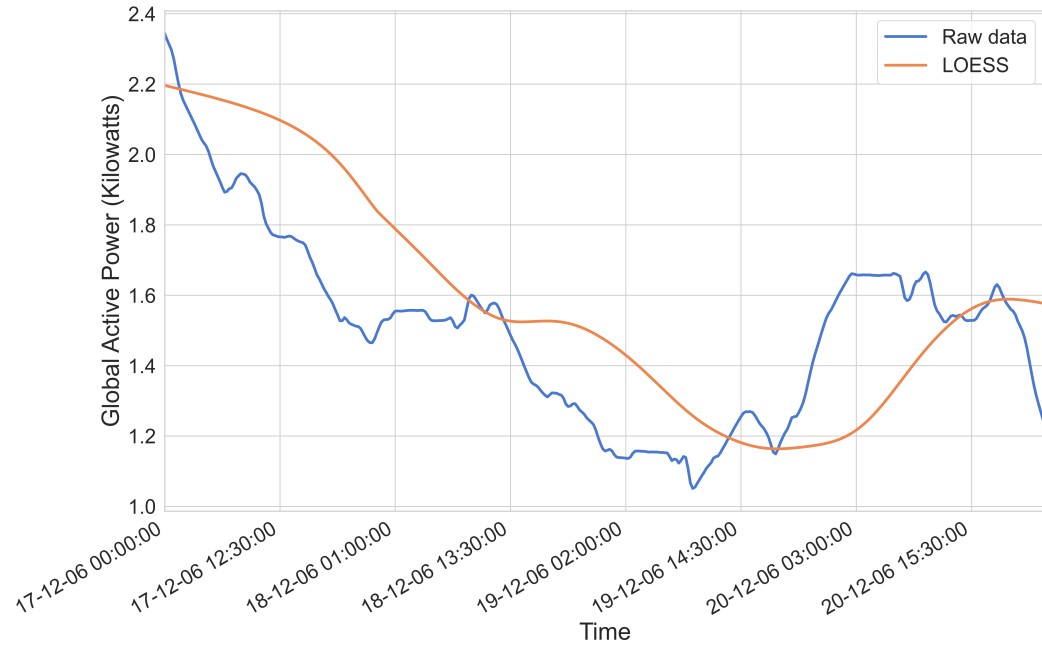

**Figure 9.** An illustration of the previously obtained trend component both with and without the application of LOESS.

The final step taken as part of Stage 3 of the forecasting pipeline was to split the data into 3 subsets that served to act as training, validation, and testing sets that were fed to our classification tree and the CNN-LSTM network that we used for the purpose of forecasting. A split employing an arbitrarily selected ratio of 60:20:20 was taken. Given the nature of our study, we chose not to shuffle the data in either of the generated sets as we were primarily

interested in our model's capability to forecast future trends in energy consumption given a measure of historically available data.

### 4.4. Stage 4— Training and Testing

In contrast to the earlier stages, stage 4 is subdivided into Sections 4.4.1 and 4.4.2, where Section 4.4.1 serves to present an overview of our classification model, while Section 4.4.2 serves to present an overview of our forecasting model.

#### 4.4.1. Stage 4.1— Classification Tree

Before attempting to forecast trends in energy consumption, we needed to establish, or otherwise ascertain, our ability to correctly assign a new point (or day) to the *correct* cluster. Given that the previously discussed clustering step separated the days in our dataset on the basis of similarity in terms of patterns in energy consumption, it was predicted that this would not be an easy feat, as it was predicted that the remaining available context information may not suffice to provide relevant information to draw up a decision boundary (of sorts) that serves to differentiate individual clusters.

The first step toward ensuring that a decently trained classifier had been developed was to deal with the glaring problem of class imbalance. The results of our clustering step led us to an uneven distribution of days among the different class labels, which could have led to poor predictive performance, as standard classification algorithms are inherently biased to the majority class. A common means to alleviate this issue is to either undersample the majority class(es) or oversample the minority class(es). In this study, we utilized the SMOTE algorithm [27], a form of informed oversampling, which works on the basis of generalizing the decision region for minority classes and thus provides us with synthetic samples while preventing overfitting. For further explanations as to the workings, advantages, and shortcomings of this algorithm, we refer the reader to the initial paper by Chawla et al. [27] as well as Figure 10, which provides a layman's explanation of the algorithm. The results of applying the SMOTE algorithm and the overall negation of the previously mentioned class imbalance can be seen in Figure 11.

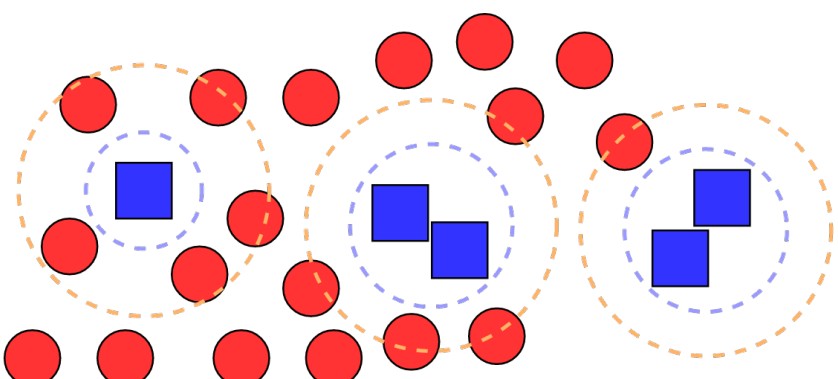

No neighbors of the same class - **Noise.**     Surrounded by another class - **Potentially unsafe.**     Minimal neighbors from other class - **Safe.**

**Figure 10.** An illustration of the Synthetic Minority Oversampling Technique (SMOTE) algorithm in the case of 2 classes depicted by blue squares (minority class) and red circles (majority class). The blue square on the far left is isolated from other members of its class and is surrounded by members of the other class; this is considered to be a noise point. The cluster in the center contains several blue squares surrounded by members from the other class and thus is indicative of potentially *unsafe* points that are unlikely to be random noise. Finally, the cluster in the far right contains predominantly isolated blue squares. The algorithm will then generate new synthetic samples, prioritizing the safer regions.

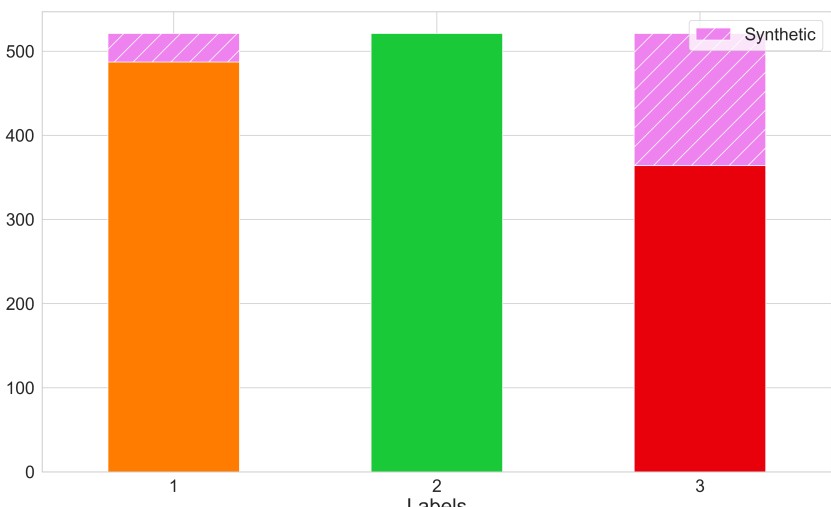

**Figure 11.** Number of samples per class label after applying the SMOTE algorithm.

After handling the class imbalance problem, we shifted our attention to feature engineering and the feature selection process of this particular classification problem. In this scenario, the available context information that we had was purely temporal (ordinal day of the week/year, month, season, etc.) and historical as well as forecasted meteorological data (air temperature, humidity, cloud opacity, etc.), and this served to act as the baseline features received by our classifier and used to assign new samples into the previously generated clusters.

As part of our research, we made use of a Random Forest Classifier, the hyperparameters of which were tuned through a randomized search over a predetermined distribution of values for each hyperparameter. After assessing the optimal hyperparameters for our use-case, we passed the model as well as the complete set of features through a feature selection algorithm titled Recursive Feature Elimination and Cross-Validation (RFECV), which works by providing a cross-validated selection of the most important features when considering a target label and pruning the less important features. Application of this algorithm reduced the overall number of features that our Random Forest Classifier utilized from an initial 77 down to a mere 24 (as shown in Figure 12)—an overall reduction of $\sim$ 68%, which was predicted to lead to better predictive and run-time performances.

After transforming the dataset and pruning the less important features we, once again, trained the model on the new transformed dataset utilizing 5-fold stratified cross-validation to assess whether our model was overfitting at any stage and to ensure an even distribution of class labels for each validation set. We conducted a quick inspection of the now fitted model by calculating the permutation feature importance on a per-feature basis to validate whether the final set of features was relevant when attempting to classify a new sample into the correct cluster. By definition, the permutation importance of a feature is the overall decrease in accuracy of our model when the said feature's values are randomly shuffled. By implementing this concept, we broke the relationship between the feature and the target label and were able to assess the dependence of our model on that feature, the results of which can be seen in Figure 13. It is important to note that, when calculating the permutation importance of strongly correlated features, the model will still have access to the shuffled feature through its correlated feature, which will result in lower importance values for *both* features when they might actually be important. To address this, we pruned the data set and removed subsets of features with strong inter-correlation.

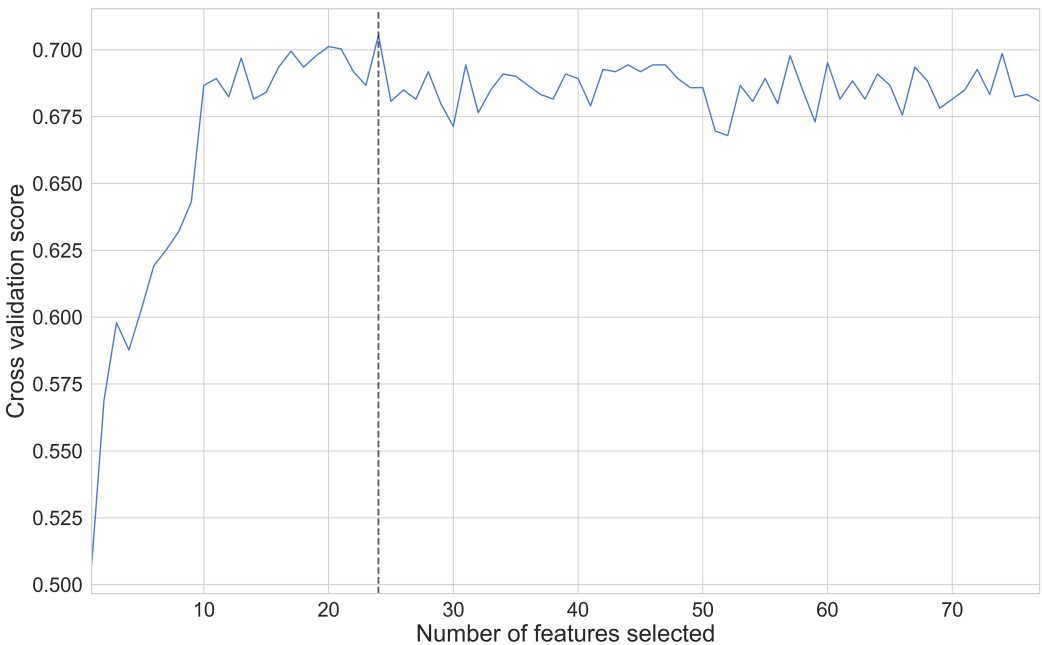

**Figure 12.** Assessing the number of important features through the use of the Recursive Feature Elimination and Cross-Validation (RFECV) algorithm. In this particular scenario, the optimal number of features was pruned down from a total of 77 to a mere 24.

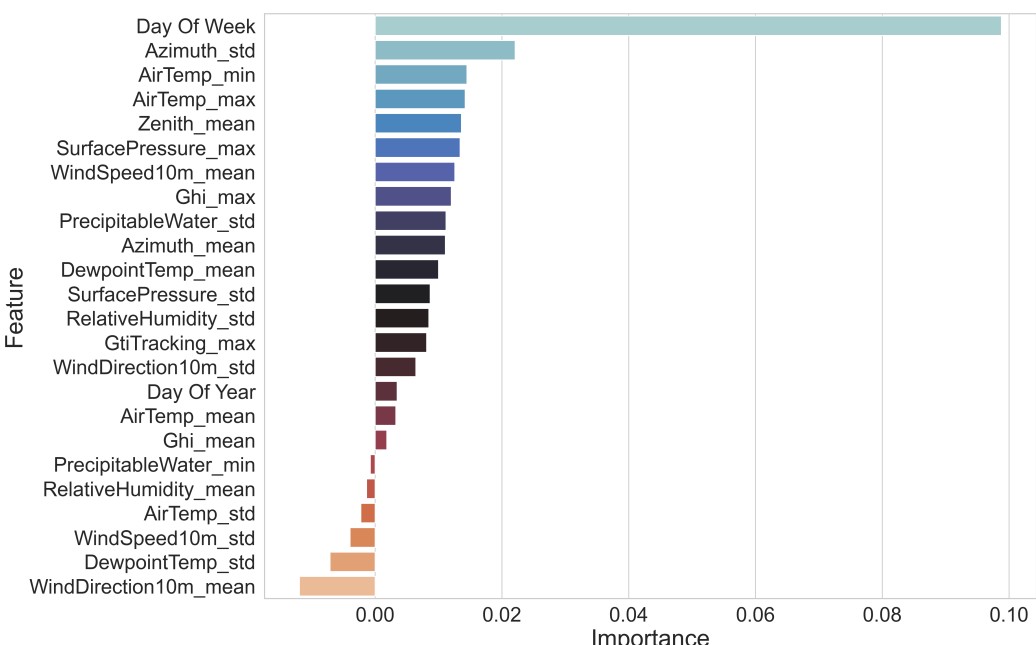

**Figure 13.** The permutation importance of each of the features chosen as part of our fitted Random Forest classifier.

The final model was then ready to accept new samples and assign them cluster labels based on the training procedure outlined through Section 4.4.1.

### 4.4.2. Stage 4.2— CNN-LSTM Network

Finally, when taking the forecasting step of our method into consideration, we chose to implement a CNN-LSTM model in which the Convolutional Neural Network (CNN) component is used to learn the relative importance of each of the features (temporal as well as meteorological) and this is passed to the network as input in what we can loosely call a

*feature extraction* step. The extracted features are then passed to the LSTM portion of the network, which learns the temporal relationships with past, or otherwise historical, values of said features with the present, or future, value(s) of the target variable, and finally, an output prediction is made. The combination of both CNN and LSTM components allows the network to learn spatio-temporal relationships between the features being passed as input and the target variable that we are attempting to forecast. In contrast to other architectures and forecasting models, this architecture is demonstrably more efficient for tackling time series problems such as those of residential energy consumption forecasting [9]. The sample network illustrated in Figure 14 can be expanded to forecast multiple time steps ahead with minor adjustments and is capable of understanding patterns at variable time resolutions. For the purposes of this example, we used the previously defined resolution of 15 min using a window of 24 historical values $(t - 24, t - 23, ..., t)$ to make a prediction one step into the future $(t + 1)$ for both the previously established trend component as well as the raw, unadulterated data.

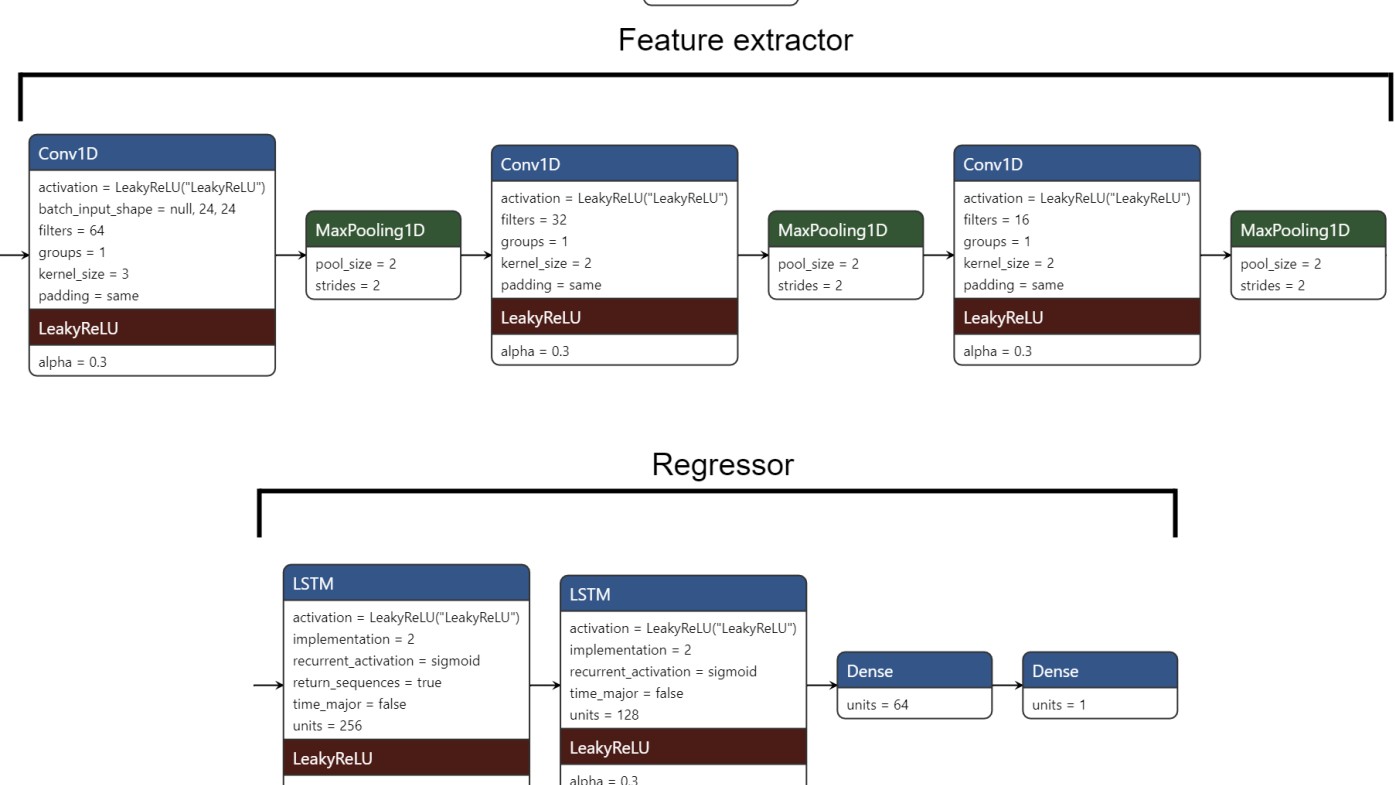

**Figure 14.** A simple, example CNN-LSTM network that makes one-step-ahead predictions.

To train our network, we utilized Adam [28]: an adaptive learning rate optimization algorithm that was designed specifically for training deep neural networks. In contrast to the ever-familiar Stochastic Gradient Descent, Adam leverages the power of adaptive learning rate methods and momentum to allocate individual learning rates for each parameter of the network being trained. For further explanations about workings of this algorithm, we refer the reader to the initial paper by Kingma and Ba [28]. Additionally, when training our network(s), we made use of a variety of techniques to improve generalization and prevent overfitting of the training data set(s). The first of these techniques was the notion of *early stopping*. Early stopping is a form of regularization that monitors the validation loss (or generalization error) and aborts training when the monitored values either begin to

degrade or do not shift for an arbitrarily set number of epochs. The second technique used worked on the notion of employing a variable learning rate which, in theory, facilitated convergence of our weight update rule and prevented learning from stagnating, thus allowing us to break through plateaus and avoid settling at local minima. For the purposes of our experiments and procuring the results showcased in Section 5, we implemented a network on a per-cluster basis for both the raw data as well as the trend component of each of our datasets. The networks implemented served to provide one-step-ahead forecasts as well as one-shot 12-step-ahead (3 h) forecasts as proof of concept.

## 5. Results and Discussion

Following the brief example in Section 4, we extended the implementation to house 12 of the REFIT dataset. The subsequent sections demonstrate the efficacy of both the classification step aand the forecasting step of our method.

### 5.1. Clustering

The first results presented are those of the clustering step of our method. We start off by presenting a scatter plot of the two-dimensional output obtained as a result of performing the UMAP algorithm on the UCID dataset, which can be seen in Figure 15. This allows us to clearly visualize the two-dimensional interpretation of the samples present in the UCID dataset. Each of the points found on the two-dimensional surface in Figure 15 represents a single day, and given that the UMAP algorithm claims to preserve both the local as well as most of the global structure present in the data, we can safely assume that distances between the samples are conducive to the similarity in terms of energy consumption as per the previously segmented interpretation of the data.

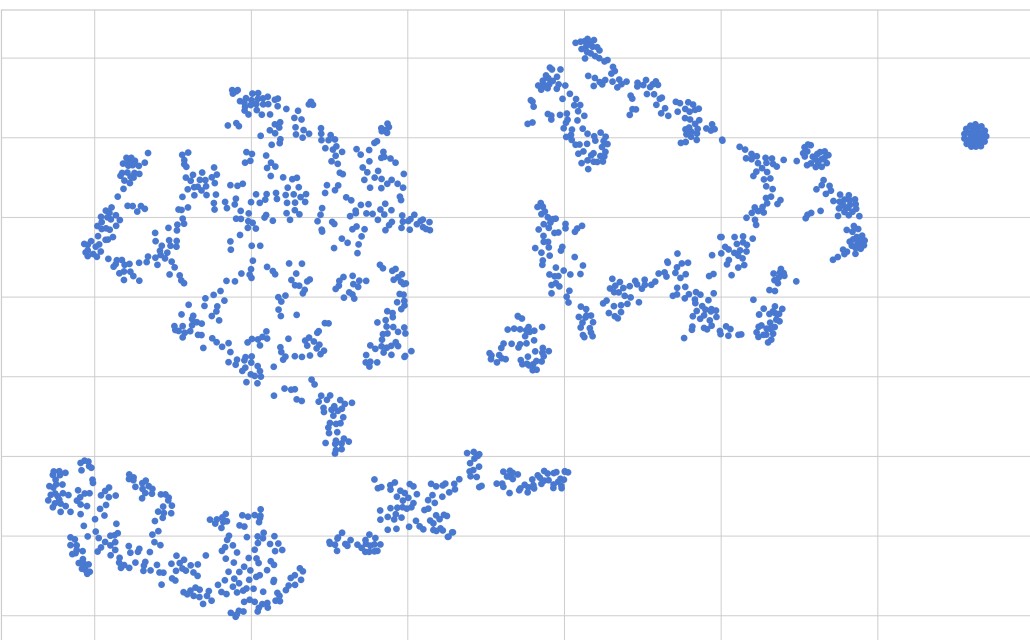

**Figure 15.** The output of performing the UMAP algorithm on the 20-dimensional UCID dataset. Each point in this figure represents a single sample (or day) within our dataset mapped onto a 2-dimensional surface.

Following this, we present the results of applying the HDBSCAN algorithm on the resulting two-dimensional output observed in Figure 15, which can be seen in Figure 16. Immediately, the HDBSCAN algorithm's potential for recognizing densely populated (as well as variably shaped) regions of samples without needing a priori knowledge about the number of clusters present in the data can be observed. In addition to this, we note the HDBSCAN algorithm's capability to recognize noise points (note the subset of points

coloured in black that do not clearly belong to any of the three generated clusters), which is a clear advantage over many other clustering algorithms. Given our previous assumption, it is safe to assume that the generated clusters visualized in Figure 16 contain samples that exhibit some measure of similarity with regards to their energy consumption patterns.

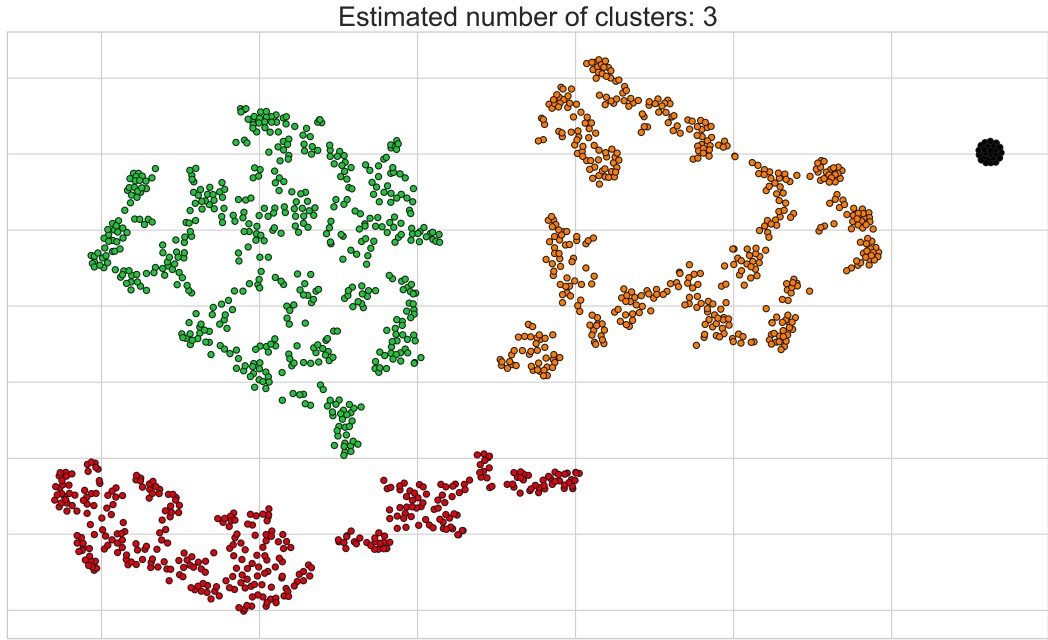

**Figure 16.** The output obtained from performing the HDBSCAN algorithm on the 2-dimensional UCID dataset previously seen in Figure 15.

For the sake of comparison, we present the output obtained by applying the k-means clustering algorithm (assuming $k = 3$) on the same two-dimensional representation of the UCID dataset (visualized in Figure 15) which can be seen in Figure 17. We immediately note the capability of the HDBSCAN algorithm to capture a better representation of the clusters present in our two-dimensional representation of the UCID dataset. The representation of outliers as noise points and not having to have a priori knowledge of the number of clusters present in the data we are working with are definite advantages that further compound our choice of clustering algorithm in our proposed model.

Visualizing, or otherwise manually inspecting, the clusters obtained as a result of our application of the HDBSCAN algorithm was necessary so that we could better understand whether our clustering algorithm can truly capture the habits of the individuals residing in the households we are working with. The first step in our analysis of the resulting clusters was to plot the averaged power consumption on a per cluster basis so that we could clearly visualize the patterns in power consumption for each cluster. An example of this, in line with the previous examples showcasing our proposed model on the UCID dataset, can be seen in Figure 18a. We note that, in this example, a subset of our data (24 samples in total) was recorded as noise by the HDBSCAN algorithm. Inspecting these samples manually led us to the confirmation that, of the 4 years worth of data, these 24 days were the only days that exhibited no tangible shifts in terms of power consumption throughout the entirety of the day (i.e., the global active power draw observed was completely stationary throughout this period); however, this is not explicitly outlined in the documentation of the UCID dataset. This can be seen as a more or less flat line in Figure 18a.

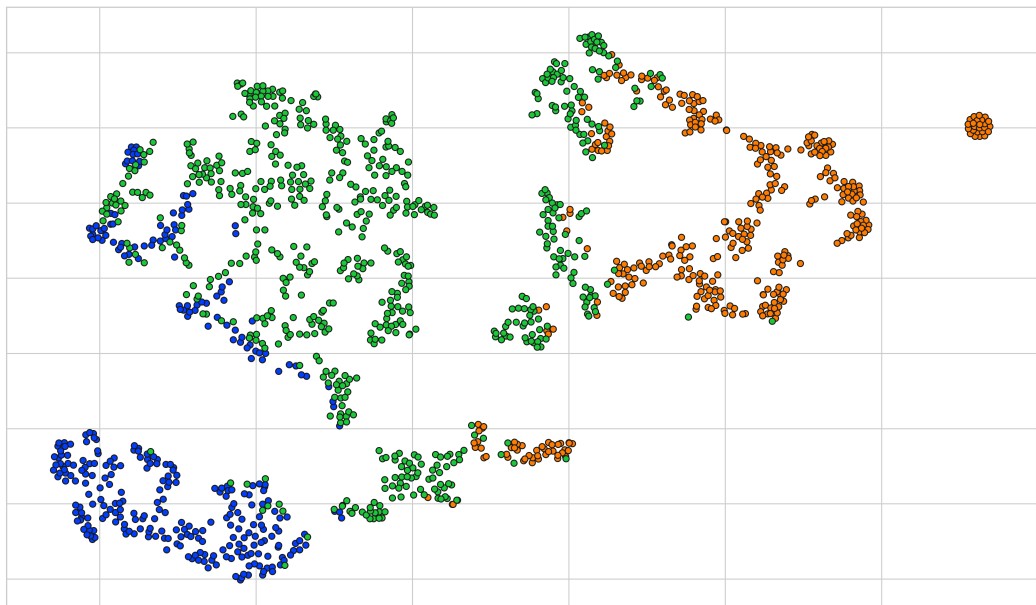

**Figure 17.** The output obtained by performing the k-means algorithm on the 2-dimensional UCID dataset previously seen in Figure 15.

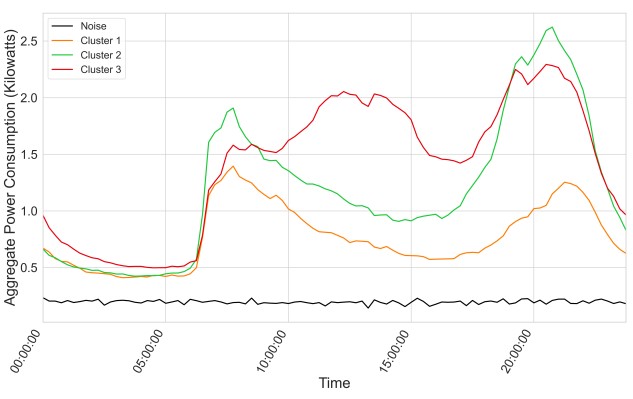

(**a**) Average power consumption per hour of the day for each of the resulting clusters obtained after utilizing the HDBSCAN algorithm on our 2-dimensional representation of the UCID dataset.

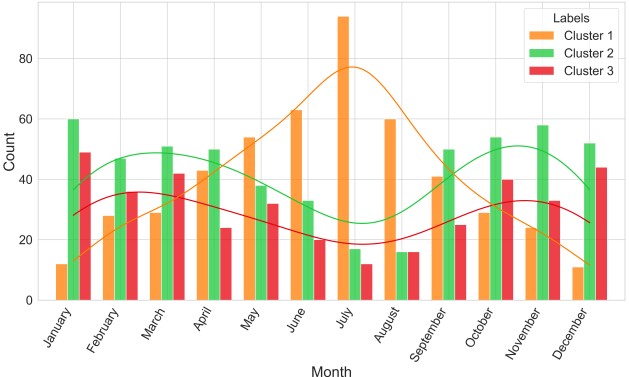

(**b**) Distribution of the clusters over the different months of the year.

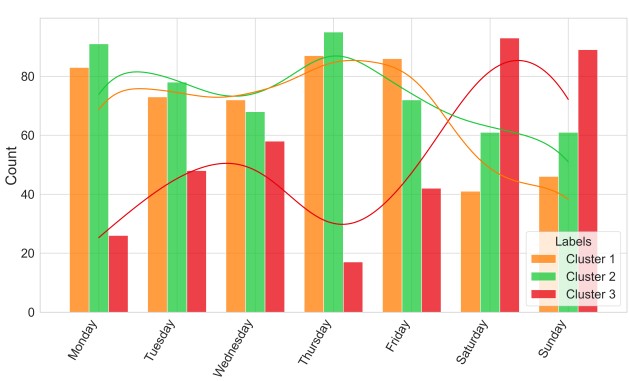

(**c**) Distribution of the clusters over the different days of the week.

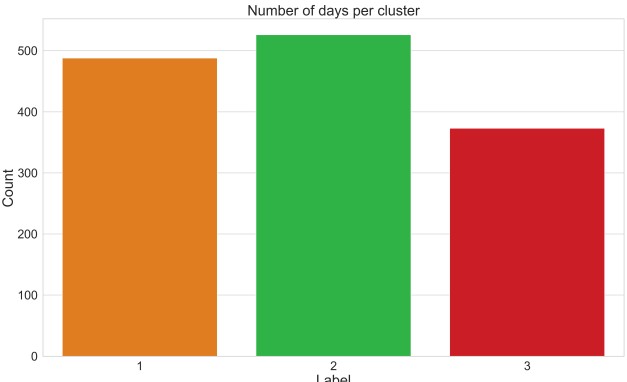

(**d**) Spread of the number of samples per cluster label.

**Figure 18.** Visualization of the generated clusters.

Figure 18b,c helps us visualize the distribution of the clusters over the months of the year as well as the days of the week to ascertain whether any of the clusters present any correlations with these temporal variables. Given that the initial spread of the data throughout the months of the year and days of the week of the UCID dataset was relatively uniform, we did not expect to see any bias towards any particular month or day in either Figure 18b or Figure 18c respectively. At a glance, we notice that clusters 1 and 2 were more likely to occur on weekdays, with cluster 3 taking over the majority share of the weekend, which tends to explain the more consistent draw in power throughout the entirety of the day for samples belonging to cluster 3. Furthermore, samples in cluster 1 tended to gravitate towards the warmer summer months, peaking in terms of number of occurrences in the month of July, while samples in clusters 2 and 3 exhibited more uniform spread over the remainder of the colder months. This could explain the lower average draw in power present in samples belonging to cluster 1 as a result of the owners of the home not being in it as often or potentially not needing to make use of appliances to heat up their home (we note that this data was collected in Sceaux, France which experiences a warm season of $\sim 3$ months with otherwise generally cooler temperatures).

N.B.: It is worth noting that the performance of these same steps on households from within the REFIT dataset exhibit similar results.

### 5.2. Cluster Label Classification

The second results to be presented are those of the classification step of our method (refer to Table 3).

**Table 3.** Results of training, optimizing, and evaluating a random forest classifier on the cluster labels obtained for the UCID and REFIT datasets.

| Data Set | No. of Clusters | Accuracy |
|---|---|---|
| UCID | 3 | 76% |
| REFIT - House 12 | 3 | 66% |

Being able to correctly assign new samples to the correct cluster is imperative so as to ensure the highest likelihood of achieving a consistently reliable forecasting accuracy. Given that we had an equal number of three clusters per dataset and that we were working with a (synthetic) uniform distribution of samples over the different clusters, the scores outlined in Table 3 are fairly good (a random predictor would achieve an accuracy of 33.3%). The disparity in the results between the two datasets could predominantly be linked to the following two reasons:

1. The UCID dataset contained a much larger number of samples (days).
2. The distribution of the samples over the different days of the week and months was much more uniform in the UCID dataset.

Figure 19a,b allows us to clearly visualize both the correct and the incorrect predictions made by our model. Interestingly, given that both the clusters formed for the UCID and REFIT datasets were quite similar in terms of the overall patterns that were captured, the model fitted for each dataset seems to have made mistakes or incorrect predictions of a similar magnitude, with cluster 2 containing the largest number of incorrect predictions for each of the datasets and cluster 1 containing the largest number of correct predictions.

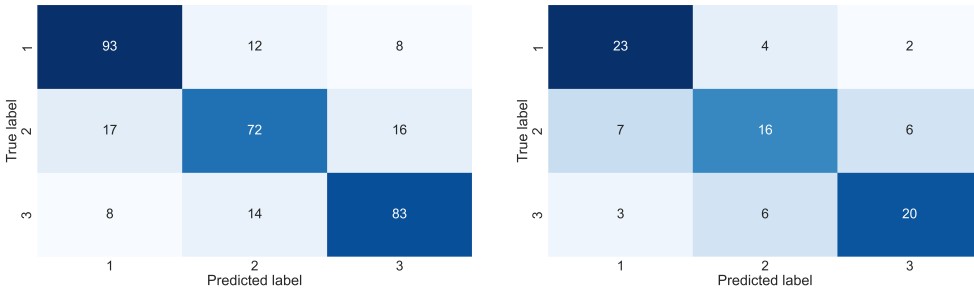

(**a**) Confusion matrix - UCID.  (**b**) Confusion matrix - REFIT.

**Figure 19.** Confusion matrices for the REFIT dataset and the UCID dataset.

### 5.3. Forecasting Accuracy

Compared with the current state-of-the-art methods presented in modern literature, particularly with regard to data available that pertain to the UCID data set, our method yields superior forecasting accuracy at variable resolutions. Table 4 presents a performance comparison of common models discussed in the literature and our method for forecasting one step into the future. We note that, at the time of writing, no published studies have attempted to forecast energy consumption on the REFIT dataset, and thus, rather than attempting to recreate the results ourselves, we decided to omit them from Table 4 for the time being.

**Table 4.** Performance comparison of different methods used for the UCID and REFIT datasets. Note that these results were obtained for one-step-ahead predictions at a resolution of 15 min from the raw datasets.

| Data Set | Method | MAE (kW) | RMSE (kW) | MAPE |
|---|---|---|---|---|
| | LSTM [9] | 0.62 | 0.86 | 51.45% |
| UCID | CNN-LSTM [9] | 0.34 | 0.61 | 34.84% |
| | Proposed | 0.14 | 0.19 | 21.62% |
| | LSTM | N/A | N/A | N/A |
| REFIT | CNN-LSTM | N/A | N/A | N/A |
| | Proposed | 0.11 | 0.17 | 25.77% |

Another component that is frequently (attempted to be) forecasted in the literature is the trend component obtained as part of a time-series decomposition step. We attempted to tackle this problem ourselves and applied the method to both the smoothed trend component of the UCID dataset as well as house 12 of the REFIT dataset, the results of which can be seen in Table 5. We note that the results were good, with a MAPE value of ~ 4% achieved for both datasets when forecasting a single time step into the future.

**Table 5.** Performance metrics obtained when applying our method on the trend component of the UCID and the REFIT datasets to obtain one-step-ahead predictions.

| Data Set | MAE (kW) | RMSE (kW) | MAPE |
|---|---|---|---|
| UCID | 0.02 | 0.02 | 2.58% |
| REFIT | 0.02 | 0.02 | 4.32% |

Finally, we attempted to extend our model by scaling up the number of predictions from a singular step (15 min into the future in this scenario) to a total of 12 sequential steps (leading to a grand total of 3 h being forecasted given the previously mentioned step size of 15 min), the results of which can be seen in Table 6.

**Table 6.** Performance metrics obtained when applying our method on both the raw data as well as the trend component of the UCID and REFIT datasets to obtain twelve-step-ahead predictions.

| Data Set | Method | MAE (kW) | RMSE (kW) | MAPE |
|----------|--------|----------|-----------|------|
| UCID | Raw | 0.37 | 0.59 | 38.23% |
|      | Trend | 0.02 | 0.02 | 3.15% |
| REFIT | Raw | 0.17 | 0.31 | 39.75% |
|       | Trend | 0.02 | 0.02 | 4.75% |

Oddly enough, for both the UCID dataset and house 12 of the REFIT dataset, we achieved marginal improvements with regard to MAPE scores when attempting to build twelve-step-ahead forecasts on the respective trend components. On the other hand, MAPE scores for the raw data for each of our datasets fell somewhat substantially, with an overall loss of about $\sim 10\%$ when moving from one-step-ahead forecasts to twelve-step-ahead forecasts, which is more in line with what one could expect in this scenario.

To further showcase or visualize the capabilities of our model, we present Figures 20a,b and 21a,b, which serve to illustrate one-step-ahead forecasts generated for a subset of the UCID dataset and house 12 of the REFIT dataset. These figures illustrate predictions made by each of our individual models on both the raw data as well as the trend component for the REFIT dataset and UCID dataset over a period of 1 day from our test set. A cursory glance at Figures 20b and 21b shows us that our model seems to excel at making one-step predictions on the trend component of the UCID dataset, while the predictions being made for the trend component of the REFIT data set seem to be slightly less accurate. When considering the raw data for each of the datasets (Figures 20a and 21a), the differences are less pronounced, and it seems that the model is capable of making accurate predictions one step into the future. This might vary between days though and, given that the days chosen for these illustrations were completely random, it might be the case that the model performs better (or worse).

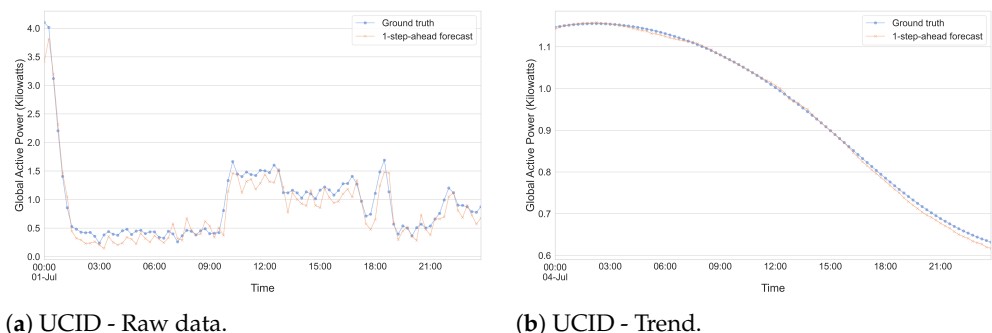

(**a**) UCID - Raw data.                                   (**b**) UCID - Trend.

**Figure 20.** Showcasing the ability of our method to make one-step-ahead predictions on the UCID dataset.

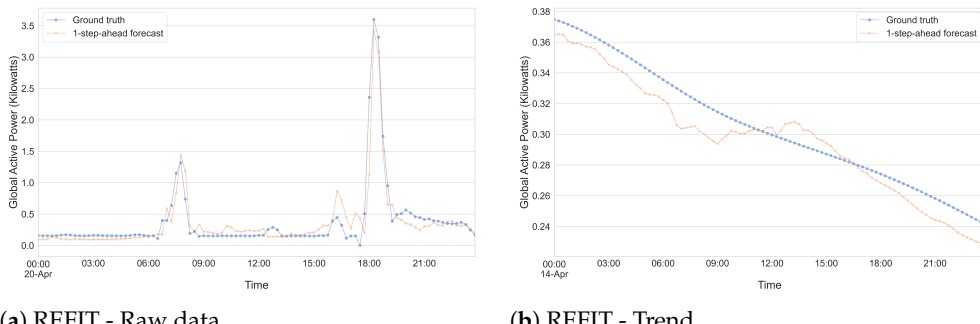

(**a**) REFIT - Raw data.                    (**b**) REFIT - Trend.

**Figure 21.** Showcasing the ability of our method to make one-step-ahead predictions on the REFIT dataset.

N.B. we note that the results obtained as part of Section 5.3 are the averaged results obtained from training, optimizing, and assessing multiple models, one for each of the respective clusters obtained as part of stage 2 of our method. Furthermore, all results were obtained at a resampled resolution of 15 min per timestep; however, similar results have been observed for variable time resolutions (1 min, 1 h, etc.)

## 6. Conclusions and Future Work

In this study, we showed that the application of a clustering step that utilizes dimensionality reduction techniques, such as t-SNE, and hierarchical density-based clustering in the form of HDBSCAN leads to significant improvements in forecasting accuracy when taking individual households into consideration. While this technique is certainly more complex, in particular with regard to the number of steps and moving parts associated with the entire pipeline, we maintain that the benefits in terms of the improved forecasting accuracy outweigh the overall increases in time and effort required to train and set up such a model. The practicality of the model lies in the availability of the data that it requires to function, primarily with respect to historical energy consumption data for the individual households in question (which is becoming easier and easier to obtain thanks to the prevalence of smart meters) and meteorological data, which can easily be obtained from numerous sources. Furthermore, it is highly likely that, given enough historical data, the need to further train the model(s) after the initial setup willb be rather low, further compounding the efficacy of our method.

Another previously discussed benefit of our method is that no prior knowledge of the number of clusters is required. As there is no guarantee that any two individual households contain similar numbers of *repeating* patterns, we avoid running into the problem of overly generalizing a single working solution that may or may not work given a change in energy consumption patterns and, instead, present a solution that could potentially extend to a much larger scale. A potential issue with this implementation, however, is that an individual household *may* contain a large number of repeating consumption patterns, which could possibly lead to an overall decline in what can already be considered a sub-par performance from our classifier. That said, there is definitely room for improvement to accommodate these potential risks, specifically with regard to the feature engineering step, for example, improvements in the classifier accuracy could be seen through the utilization of a more efficient classifier. Alternatively, the current lack of contextual information that serves to explain the emergence of the clusters as part of the clustering step could be the reason for obtaining sub-par accuracy scores as, in its current iteration, the premise of our clustering step was to group together days that exhibited the highest level of similarity purely in terms of their energy consumption patterns and, given that this information is not readily available to us when considering a new day, we were left reaching for straws when attempting to explain when any individual household was likely to observe energy consumption patterns falling within any of the obtained clusters. Evidently, temporal and meteorological information is not enough to explain the emergence of said clusters, and

other information (perhaps patterns in terms of cluster labels leading up to the new sample) could serve to improve the classifier accuracy. This is definitely an area of this study that could be looked into as part of future research. Additionally, regardless of the fact that the performance of our forecasting model is the highlight of this paper, it is interesting to note that a byproduct of our method is the potential to extract insights into variables that affect the daily energy consumption patterns of unique households. A cursory glance at applying our method to a portion of the data at hand, as an example of the insights that can be obtained, shows us that some households have frequently occurring patterns that tend to deviate among the different days of the week while other households have an even bigger separation across months of the year or even among meteorological factors such as the temperature or chance of rain.

Additionally, given the previously described practical application within a HEMS or BEMS setup, our method should allow significant improvements over the current state-of-the-art methods in terms of improving solutions that are built on the smart grid framework. Given the greater complexity of our method compared with other available methods, it is worth noting that an increase in training time is inevitable; however, as previously stated, this should be a nonissue given the high availability of historical energy consumption data. The practicality of the method in terms of data requirements is also a nonissue, as the only data required are the available smart meter data as well as other variables that are easy to obtain and otherwise publicly available. Another significant advantage of this method is the insights drawn as a result of obtaining energy profiles unique to each individual household that would not be readily available without some manual work (usually in the form of surveys or otherwise) by local energy companies. This method could also potentially see extensions that go beyond energy consumption forecasting (assuming that the data on hand comprises a similar structure). To conclude, we note that, as a result of preclustering our data utilizing a hierarchical density-based clustering algorithm in the form of HDBSCAN and then training separate CNN-LSTM models on a per-cluster basis, we achieved an improvement in overall forecasting accuracy with superior MAPE scores (21.62% in contrast to 34.84%) when considering the current state-of-the-art methods (LSTM networks, clustering based on K-means, etc.), and ew were also able to form unique energy profiles for individual households, providing valuable insights into their respective energy consumption habits as well as the factors influencing their energy consumption.

**Author Contributions:** K.A.-S., the main author, implemented the software, performed the validation, analysis, and writing. This project was completed as his Master Thesis. Both V.D. and M.M. were involved as supervisors of K.A.-S., and in the methodology analysis and validation, with V.D. being additionally involved in project conceptualization. All authors have read and agreed to the published version of the manuscript.

**Funding:** This research received no external funding.

**Institutional Review Board Statement:** Not applicable.

**Informed Consent Statement:** Not applicable.

**Data Availability Statement:** The data used in this paper are publicly available [10,21]. In the interest of open science, the code has also been made publicly available https://github.com/rug-ds-lab/msc-thesis-s3877043-al-saudi-kareem, accessed on 29 August 2021. Any use must include a citation to this paper.

**Conflicts of Interest:** The authors declare no conflict of interest.

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
