# Peer review of "Energy Consumption Patterns and Load Forecasting with Profiled CNN-LSTM Networks"

_processes, doi:10.3390/pr9111870_

Round 1
Reviewer 1 Report
This paper proposes a novel load forecasting method that utilizes a clustering step prior to the forecasting step to group together days that exhibit similar patterns in energy consumption. Following that, the authors attempt to classify new days into one of the pre-generated clusters by making use of the available context information (day of the week, month, predicted eather). Finally, using available historical data (with regards to energy consumption) alongside meteorological and temporal variables, we train a CNN-LSTM model on a per-cluster basis that each specializes in forecasting based on the energy profiles present within each cluster. To further improve the quality of this paper,I have the following concerns:
1)The main contributions should be summarized in introduction part;
2)The influence factors of energy consumption should be discussed;
3)The cluster process in your work should be more detailed.
Author Response
1)The main contributions should be summarized in introduction part;
Section 3 (originally the proposed model) has been moved to the introduction and further elaboration as to the main contributions of the paper have been appended to that subsection.
2)The influence factors of energy consumption should be discussed;
In the results and discussion section under the clustering subsection - visualizations as well as influence factors of energy consumption are discussed. This is also touched upon in the conclusion section of the paper.
3)The cluster process in your work should be more detailed
An explanation (as well as citation) has been provided with respect to our clustering algorithm of choice (HDBSCAN).
Reviewer 2 Report
The paper proposes a new method for predicting load and energy consumption patterns, based on neural networks.
The abstract provides a well described overview of the work, the method presented and the results obtained.
In the introduction is presented a survey of the literature associated with the problem addressed, about prediction of energy consumption patterns. However, I missed a more explicit description of both the general objective and the contributions, which may lead readers to be more interested in the content of the other sections of the paper.
In the second section, the authors describe all the datasets considered in their analysis and scenarios studied. Imagining that the intention was to describe the methodological aspects of the work, I missed the link to the referent session, which only appears in session 5. My suggestion here is that this materials and methods session be brought forward and the description about the data sets be brought into it. I also suggest improving figure 3.
In the third session, the authors present the proposed model. I point out the need for improvement in the quality of figure 1, which describes the block diagram of the model, which is described in a limited way in the session, as well as a more detailed description of the proposed model, preferably with a description of the algorithms.
The fourth section presents the related papers. For the purpose of facilitating the flow of the text, this section would be better used if it were closer to the first session, so that it is easier to build and highlight the research gap that the paper attacks, precisely to highlight more sharply the contribution of the paper to the topic.
Session 5 is a bit confused. Although it is called "Materials and Methods", it actually contains several results of the paper mixed with the description of the methods. My suggestion here, to organize the text, would be, perhaps, to use the information that describes how the work will be done (basically explaining the blocks in figure 3), within session 2, which would become the "Materials and Methods" session. This would probably make the text clearer to the reader. The other informations in the session, such as the results presented in most of the figures already present results and are discussed as such, making no sense to be in a session of materials and methods, and should therefore be added to the next session, "Results and Discussion".
In session 6, the authors present other examples besides the one discussed in session 5, in order to consider different scenarios to study the efficiency of the proposed method. Naturally, the session leads the reader to consider the example worked in session 5, justifying and explaining why this example has to be placed in this session and not in "Materials and Methods". In this session, one could even work on a direct comparison between the other methods considered using other metrics besides accuracy, summarized in table 4 of the text. And, possibly, highlight scenarios of advantages and disadvantages of using the method proposed in this paper when compared with the other techniques considered.
Finally, the conclusions are presented very briefly, but sufficiently to discuss the results presented. I believe it would be interesting to stress the question of which scenarios are most interesting for the use of the proposed method, at least as a basis for future investigations.
Author Response
In the introduction is presented a survey of the literature associated with the problem addressed, about prediction of energy consumption patterns. However, I missed a more explicit description of both the general objective and the contributions, which may lead readers to be more interested in the content of the other sections of the paper.
Section 3, previously elaborating on our proposed model, has been shifted to the introduction where we have elaborated on the general objective as well as the contributions of the paper.
In the second section, the authors describe all the datasets considered in their analysis and scenarios studied. Imagining that the intention was to describe the methodological aspects of the work, I missed the link to the referent session, which only appears in session 5. My suggestion here is that this materials and methods session be brought forward and the description about the data sets be brought into it. I also suggest improving figure 3.
We have renamed the section to "Data Set Description" - it is now also section 3 of the paper coming directly before section 4 (the referrent section) which has been renamed from "Materials and Methods" to "Methodology". The clarity of figure 3 has been improved.
In the third session, the authors present the proposed model. I point out the need for improvement in the quality of figure 1, which describes the block diagram of the model, which is described in a limited way in the session, as well as a more detailed description of the proposed model, preferably with a description of the algorithms.
The clarity of figure 1 has been improved. A description of the algorithms would drastically increase the length of the paper. Each algorithm is instead cited and, when required, a short description is provided.
The fourth section presents the related papers. For the purpose of facilitating the flow of the text, this section would be better used if it were closer to the first session, so that it is easier to build and highlight the research gap that the paper attacks, precisely to highlight more sharply the contribution of the paper to the topic.
Section 4 has been moved to the beginning of the paper and now comes directly after the introduction.
Session 5 is a bit confused. Although it is called "Materials and Methods", it actually contains several results of the paper mixed with the description of the methods. My suggestion here, to organize the text, would be, perhaps, to use the information that describes how the work will be done (basically explaining the blocks in figure 3), within session 2, which would become the "Materials and Methods" session. This would probably make the text clearer to the reader.
The other informations in the session, such as the results presented in most of the figures already present results and are discussed as such, making no sense to be in a session of materials and methods, and should therefore be added to the next session, "Results and Discussion".
All results present in the (now) Methodology section have been moved to the Results and Discussion section which now contains results pertaining to each section of our method (Clustering, Classification, Forecasting).
In session 6, the authors present other examples besides the one discussed in session 5, in order to consider different scenarios to study the efficiency of the proposed method. Naturally, the session leads the reader to consider the example worked in session 5, justifying and explaining why this example has to be placed in this session and not in "Materials and Methods". In this session, one could even work on a direct comparison between the other methods considered using other metrics besides accuracy, summarized in table 4 of the text. And, possibly, highlight scenarios of advantages and disadvantages of using the method proposed in this paper when compared with the other techniques considered.
See previous comment. Additionally, we have expanded on the advantages/disadvantages of our proposed method in the Conclusion of our paper.
Finally, the conclusions are presented very briefly, but sufficiently to discuss the results presented. I believe it would be interesting to stress the question of which scenarios are most interesting for the use of the proposed method, at least as a basis for future investigations.
The conclusion has been expanded and now include scenarios in which the proposed method would be most useful as well as use-cases outside of energy consumption as well as the previously mentioned advantages/disadvantages.
Round 2
Reviewer 1 Report
The authors have solved the problems supplied by the author, and I think that the revised paper can be accepted in its current form.